# Research on Obstacle-Avoidance Trajectory Planning for Drill and Anchor Materials Handling by a Mechanical Arm on a Coal Mine Drilling and Anchoring Robot

**DOI:** 10.3390/s24216866

**Published:** 2024-10-25

**Authors:** Siya Sun, Sirui Mao, Xusheng Xue, Chuanwei Wang, Hongwei Ma, Yifeng Guo, Haining Yuan, Hao Su

**Affiliations:** 1College of Mechanical Engineering, Xi’an University of Science and Technology, Xi’an 710054, China; sunsiya412@xust.edu.cn (S.S.); xuexsh@xust.edu.cn (X.X.); wangchuanwei228@xust.edu.cn (C.W.); guoyifeng000721@163.com (Y.G.); suh@stu.xust.edu.cn (H.S.); 2Shaanxi Key Laboratory of Mine Electromechanical Equipment Intelligent Detection and Control, Xi’an 710054, China; 3College of Electrical and Control Engineering, Xi’an University of Science and Technology, Xi’an 710054, China; 24206204126@stu.xust.edu.cn (S.M.); 21406050426@stu.xust.edu.cn (H.Y.)

**Keywords:** anchor drilling robot, material handling manipulator arm, dynamic obstacle avoidance, trajectory planning

## Abstract

At present, China’s coal mine permanent tunneling support commonly uses mechanized drilling and anchoring equipment; there are low support efficiency, labor intensity, and other issues. In order to further improve the support efficiency and liberate productivity, this paper further researches the trajectory planning of the drilling and anchoring materials of the robotic arm for the drilling machine “grasping–carrying–loading–unloading” on the basis of the drilling and anchoring robotic system designed by the team in the previous stage. Firstly, the kinematic model of the robotic arm with material was established by improving the D-H parameter method. Then, the working space of the robotic arm with the material was analyzed using the Monte Carlo method. The singular bit-shaped region of the robotic arm was restricted, and obstacles were removed from the working space. The inverse kinematics was utilized to solve the feasible domain of the robotic arm with material. Secondly, in order to avoid blind searching, the guidance of the Bi-RRT algorithm was improved by adding the target guidance factor, and the two-way tree connection strategy for determining the feasible domain was combined with the Bi-RRT algorithm’s feasible domain judgment bi-directional tree connection strategy to improve the convergence speed of the Bi-RRT algorithm. Then, in order to adapt to the dynamic environment and avoid the global planning algorithm from falling into the local minima, on the basis of the above planning methods, an improved Bi-RRT trajectory planning algorithm incorporating the artificial potential field was proposed, which takes the planned paths as the guiding potential field of the artificial potential field to make full use of the global information and avoid falling into the local minimization. Finally, a simulation environment was built in a ROS environment to compare and analyze the planning effect of different algorithms. The simulation results showed that the improved Bi-RRT trajectory planning algorithm incorporating the artificial potential field improved the optimization speed by 69.8% and shortened the trajectory length by 46.6% compared with the traditional RRT algorithm.

## 1. Introduction

According to statistics, the support time in a tunneling cycle operation in a coal mine roadway accounts for about 2/3 of the total time. Therefore, improving the intelligence of the permanent support equipment in the roadway is an important way to improve the efficiency of tunneling. Since 2015, the state ministries and commissions have successively issued some technical guidelines to enhance the level of coal mine development and improve the level of safety production, which is of great significance for the transformation of the permanent support equipment of the underground roadway of coal mines to intelligence [1,2,3]. Automatic loading and unloading drilling and anchoring material technology is one of the key technologies in the research of intelligent drilling and anchoring platforms [4]. The common coal mine underground intelligent drilling anchor equipment takes a load-sensitive electro-hydraulic proportional multi-way valve as the main core, PLC as the core control unit, remote control as the operation terminal, and displacement sensor, pressure sensor, and so on, as the detecting and feedback elements [5,6,7,8]. A few drilling anchor platforms have multiple automatic loading and unloading drilling rod mechanisms, and their intelligent cooperative control part is still in the research stage. The existing equipment has problems such as complex structure, many detection elements, low positioning accuracy, large repetitive positioning errors, and a single loading and unloading material, which has a large impact on the reliability of support operations [9,10,11]. In this paper, on the basis of intelligent drilling and an anchoring robot structure with a multi-degree-of-freedom robotic arm proposed by the team in the previous stage, we study a method of obstacle avoidance trajectory planning for a robotic arm with material in a dynamic environment.

For the problem of robotic arm obstacle avoidance trajectory planning, Chen et al. [12] combined non-obstacle spatial sampling with the artificial potential field method to generate heuristic sampling points, which enabled the RRT trajectory planning algorithm to obtain higher efficiency and better trajectories. Xiao et al. [13] carried out video detection by a flexible manipulator along a continuous path in a narrow 3D space and generated the trajectory under a variety of constraints, such as the terminal camera attitude, drive space, and obstacles, to generate trajectories. Zhang et al. [14] proposed a path construction strategy to remove redundant nodes to improve the smoothness and reduce the memory storage of the algorithm based on an artificial potential field method and a two-way fast exploratory random tree path planning algorithm for solving the problem of high randomness and low search efficiency. Wang et al. [15] proposed a fusion algorithm of improved artificial potential field method (APF) and rapid extended random tree method (RRT) to address the problems of low success rate and low efficiency in obstacle avoidance path planning for 6R robotic arms in complex environments. The improved APF-RRT algorithm can adapt to complex environments and effectively solves the problems of unreachable APF targets and local minima. Chen et al. [16] proposed a hybrid artificial potential field and ant colony optimization based on raster maps for solving the path planning problem of robots in known environments. Chen et al. [17] proposed a Biased-RRT correction algorithm based on sampling rule goal-oriented design and parent node re-selection to solve the path planning problem of a six-degree-of-freedom robotic arm in a complex environment. This algorithm can effectively shorten the path planning time, reduce the path length, and better accomplish the expected path planning task of a six-degree-of-freedom robotic arm in a complex environment. Zhang et al. [18] designed a Bi-RRT algorithm guided by the artificial potential field method for surface unmanned boats, and the experiment proved that the algorithm can greatly reduce the path length and the number of nodes and improve the path smoothness while improving the search efficiency.

In summary, most of the existing robotic arm obstacle avoidance trajectory optimization methods consider the end material as a mass point, and there is little research on obstacle avoidance optimization for grasping rod materials with a large aspect ratio. In the coal mine tunnel drilling and anchoring support process, the drilling and anchoring materials are mainly long and thin rods, and it is difficult to plan the trajectory of the robotic arm with the material in the restricted space-time.

## 2. Design of the Anchoring Robot System

By analyzing the process of parallel cyclic operation of coal mine tunneling and permanent support, the team designed a drilling and anchoring robot system with a collaborative robotic arm in the early stage. The system consists of an anchor drilling platform, an automatic drilling machine, a multi-degree-of-freedom collaborative robotic arm, an anchor drilling material library, and an electronic control box. The structure is shown in Figure 1. Among them, the multi-degree-of-freedom collaborative robotic arm is based on the traditional six-axis robotic arm with the addition of a horizontal moving slide and a lifting platform, which is designed to be able to meet the workspace requirements of the anchor drilling task. The structure is shown in Figure 2.

## 3. Kinematic Modeling and Feasible Domain Analysis of Material-Carrying Robotic Arm

The coal mine roadway drilling and anchoring support process is relatively cumbersome. To form the machine to complete the drilling and anchoring operations, we mainly need to solve the problem of loading and unloading of drilling and anchoring materials. As the drilling anchor materials are mostly long and thin rods, the mechanical arm clamps them in the drilling and anchoring platform delivery process. In order to reduce the probability of collision, the drilling anchor materials are required to move perpendicular to the platform to minimize its duty cycle. Therefore, the robotic arm with the material described in this paper refers to the robotic arm that clamps the drill-anchor material at the end, and special constraints are made on the attitude of this robotic arm in the process of delivering the material; the structure is shown in Figure 3.

### 3.1. Kinematics Modeling of Material-Carrying Robotic Arm

In this study, the improved D-H method is used to model and analyze the connecting rod and joints of the robotic arm with material, and the rod-shaped material is regarded as the end connecting rod of the robotic arm with material. The coordinate system of the multi-degree-of-freedom collaborative robotic arm and the connecting rod of the robotic arm with the material are established, respectively, as shown in Figure 4 and Figure 5, which are the three-dimensional coordinates of different joint points, respectively.

Based on the arm as well as the actual dimensions of the drilled anchor material, the DH parameter list of the banded robotic arm can be obtained, as shown in Table 1. $a_{i-1}$ is the length of the linkage, $\alpha_{i-1}$ is the angle of torsion of the linkage, $\theta_{i}$ is the joint variable of the rotating joints, and $d_{i}$ is the offset of the linkage.

The transformation relationship between the neighboring joints of the banding robot arm can be represented by the transformation matrix *T*. The transformation relationship between the coordinate system {i} and coordinate system {i−1} can be represented by the transformation matrix Tii−1:(1)Tii−1=cθi−sθi0ai−1sθicαi−1cθicαi−1−sαi−1−disαi−1sθisαi−1cθisαi−1cαi−1dicαi−10001

According to the D-H parameter table of the robotic arm with material, the transformation matrix of the end rod material with respect to the base coordinate system is as follows:(2)$$\theta_{5}=-\theta_{7}$$
(3)$$\theta_{6}=-\theta_{8}$$

After obtaining the displacement and inclination data of the robotic arm and substituting them into the transformation matrix, the spatial attitude of the robotic arm, as well as the drilling and anchoring material, can be obtained.

### 3.2. Collision Detection of Material-Carrying Robotic Arm

The motion of the robotic arm must be taken into account in the collision detection algorithm [19], where the kinematics are modeled in order to describe the state of motion of the robotic arm and the spatial position of the linkage. The accurate position of the robotic arm linkage obtained from the positive kinematics provides the basic data for collision detection. Subsequently, shape information is obtained by enveloping the robotic arm linkage and obstacles. Finally, it is calculated to determine whether a collision occurs or not. Because the components of the robotic arm are composed of irregular geometry, it is difficult to model accurately, and the calculation is too large. However, because the robotic arm connecting rod is similar to a cylinder, it can be used to reduce the difficulty of modeling and reduce the detection time. Regular, irregular obstacles similar to a sphere can be enveloped by a sphere, while rod-shaped obstacles can be enveloped by a number of spheres with a smaller radius. The specific modeling is shown in Figure 6.

The collision detection between the robotic arm and the obstacle can be transformed into the collision detection between the enveloped sphere and the enveloped cylinder, based on which the radius of the cylinder can be superimposed into the radius of the sphere, which can be further transformed into the collision detection between the sphere and the line segments, as shown in Figure 7.

Let the cylinder that envelopes the linkage of the robotic arm have a center A($x_{1}$,$y_{1}$,$z_{1}$) at one end and a center B($x_{2},y_{2},z_{2}$) at the other end. Let them also have a radius of $r_{x}$ and a sphere with a center C at the coordinates of the point C($x_{0},y_{0},z_{0}$) and a radius of r, and carry out the superposition after the sphere has a radius of $r_{x}+r$. Finally, let there be a point on the rod D($x_{d},y_{d},z_{d}$). According to the expression of the straight line in the space, the value of each coordinate of the point can be obtained:(4)$$\left\{\begin{array}{l} x_{d}=\lambda x_{1}+(1-\lambda )x_{2}\\ y_{d}=\lambda y_{1}+(1-\lambda )y_{2}\\ z_{d}=\lambda z_{1}+(1-\lambda )z_{2} \end{array}\right.$$
where λ is a variable between [0, 1], from which the distance from point D to the center of the sphere C is obtained:(5)$$d=\sqrt{{\left(x_{d}-x_{0}\right)}^{2}+{\left(y_{d}-y_{0}\right)}^{2}+{\left(z_{d}-z_{0}\right)}^{2}}$$

In the Formula (5), there is only one variable λ, which can build a function $f(\lambda )=d^{2}$ with a minimum value of f(λ). Then, compared with the radius of the sphere, when ${d^{2}}_{\min}>{\left(r+r_{x}\right)}^{2}$, the arm of the linkage did not collide with the obstacle, and vice versa, the collision occurred. Through the collision detection between different connecting rods and obstacles, we can finally determine whether the robotic arm has collided with the obstacle.

In addition to the collision between the robotic arm and the obstacle, the collision between the connecting rod and the connecting rod cannot be ignored due to its own structural properties. According to the structure of the six-axis robotic arm, the collision between the connecting rods generally occurs between connecting rod one and connecting rod three, connecting rod one and connecting rod four, and connecting rod two and connecting rod four. The collision between the connecting rods can be simplified as the collision between the cylinders, which is judged by calculating the distance between the axes of the two cylinders, as shown in Figure 8.

The cylinders of the envelope connecting rod one have centers *M*, *N* at each end, a radius $r_{2}$, and a point *A* on the axis. The cylinders of the envelope connecting rod 3 have centers *Q*, *P*, a radius $r_{1}$, and a point *B* on the axis, The distance between *AB* is *d*. The equation for the coordinates of points *A* and *B* can be obtained:(6)$$\left\{\begin{array}{l} A=M+\lambda (N-M)\\ B=P+\mu (Q-P) \end{array}\right.$$
where λ,μ are both variables between [0, 1], and the distance between *AB* can be calculated according to the coordinates of A, B points:(7)$$d=\sqrt{{\left(x_{A}-x_{B}\right)}^{2}+{\left(y_{A}-y_{B}\right)}^{2}+{\left(z_{A}-z_{B}\right)}^{2}}$$

The variables in Equation (7) are λ,μ. The binary function $f(\lambda ,\mu )=d^{2}$ is constructed, and the ${d^{2}}_{\min}$ distance between the two axes is obtained when f(λ,μ) takes the minimum value, while the minimum value of f(λ,μ) can be obtained by the Lagrange multiplier method:(8)$$\left\{\begin{array}{l} \frac{\partial f(\lambda ,\mu )}{\partial \lambda }=0\\ \frac{\partial f(\lambda ,\mu )}{\partial \mu }=0 \end{array}\right.$$

Solve for different sets of values of λ,μ according to the conditions of Equation (8), and substitute into Equation (7) to find the final $d_{\min}$. When $d_{\min}>r_{1}+r_{2}$ it can be assumed that there is no collision between connecting rod I and connecting rod III. Similarly, by calculating whether a collision occurs between connecting rod one and connecting rod four and between connecting rod two and connecting rod four, it can be determined whether a collision occurs between connecting rods of the robotic arm. When there is no collision between the robotic arm and the obstacle or between the robotic arm’s connecting rods, it can be assumed that the robotic arm has not collided in the space.

### 3.3. Feasibility Domain Analysis of Robotic Arm with Material

In general, the workspace of a robotic arm is defined as all the locations that can be reached by the end device after it has completed all operations, i.e., the total amount of space covered by the end-effector. A random probability-based digitization method, the Monte Carlo algorithm [20], is chosen. In this paper, the range of locations that can be reached by the robotic arm with a material is called the task space.

When a robotic arm operates in the workspace, due to its own inherent nature, certain special positions or configurations will occur so that the robotic arm loses certain degrees of freedom or fails to move normally, which is also known as the robotic arm singularity avoidance problem [21]. Due to the kinematic inverse solution, singularities generally only exist under trajectory planning in Cartesian space, and planning under joint space generally does not have singularities because it does not involve inverting the speed of the robotic arm. Under the Cartesian coordinate system, different singularities can be categorized into boundary singularities and interior singularities according to different singular bit shapes, as shown in Figure 9.

In order to avoid the above situation and better solve the feasible domain of the robotic arm, the robotic arm can be restricted according to the singularity interval of the robotic arm, so that the robotic arm moves to avoid the joint angle in the range of this interval. As for the singular point in the operation of the robotic arm, it needs to be separated from the Jacobi matrix. The Jacobi determinant equal to zero is a sufficient condition for the singular point, and the singular region of the robotic arm can be solved by judging through this condition.

After restricting the robotic arm according to the singular interval, the obstacle interval can be removed on the basis of this to get the final position that meets the inherent properties of the robotic arm itself and the surrounding environment, i.e., the feasible domain of the robotic arm. After obtaining the feasible domain of the robotic arm, the planning algorithm can first determine whether the planned trajectory points are in the feasible domain to reduce the subsequent unnecessary judgment and accelerate the convergence of the algorithm. In the ROS simulation, the workspace of the robotic arm to keep the vertical attitude of the end rod is calculated, and the simulation results are shown in Figure 10. Among them, the yellow border is the workspace, the purple border is the task space, and the green border is the feasible domain.

## 4. Obstacle Avoidance Trajectory Planning Strategy for Robotic Arm with Material in Dynamic Environment

### 4.1. Bi-RRT Improvement Strategies

#### 4.1.1. Connection Judgement

Using a fixed distance as a condition for judging whether a connection is made or not can lead to a degradation of the algorithm’s performance under certain circumstances. Therefore, the algorithm in this study introduces a new connection judgment strategy. A new connection evaluation process is performed whenever a node is created, and the detailed steps are as follows: if both $T_{a}$ and $T_{b}$ are expanding a new node, the nodes generated by each are tried to be connected, and their possible routes are detected for collision. Once any obstacle is found blocking this route, the process of connecting is terminated, and the new node expansion is sought again. Only when no collision is encountered is it confirmed that the optimal solution has been found and the trajectory planning is completed, thus stopping the next expansion. The specific process is shown in Figure 11.

As there will be a large number of calculations to perform collision detection, in order to reduce the amount of computation, set a suitable starting judgment distance value ds roughly for the starting point value of the target point distance of the general. This can only occur when the distance between the two new nodes is less than the value of the above connection operation.

#### 4.1.2. Goal Oriented

Bi-RRT is the same as the RRT algorithm; although the two-way search strategy makes the algorithm converge much faster, the growth of nodes is still random and without goals, the problem of node redundancy still exists, and the growth of the two trees may occur in the opposite direction, which reduces the efficiency of the whole algorithm [22]. In this paper, for the algorithm under study, the goal-oriented strategy is invoked, so that $T_{a}$ and $T_{b}$ are expanded to the goal point and the starting point, respectively, which guides the two trees to meet and accelerates the algorithm’s convergence. At the same time, in order to avoid the algorithm losing randomness, resulting in insufficient space for searching and falling into the local optimum, set a fixed guidance factor P, each time before generating a new node to randomly generate $P_{r}$ for comparison with P. The specific guidance formula is shown in the following equation.
(9)$$X_{new}=\left\{\begin{array}{l} X_{goal}/X_{start}\\ X_{rand} \end{array}\right.\begin{array}{l} P_{r}>P\\ P_{r}<P \end{array}$$

When $P_{r}>P$, the newly generated nodes are made to point to the starting point or the goal point, depending on the tree. When $P_{r}<P$, generate new nodes by random sampling in the space. The specific process is shown in Figure 12.

#### 4.1.3. Trajectory Simplification

The above improvement ideas can greatly reduce the speed of convergence of the algorithm, but even with the goal-oriented strategy, due to the algorithm’s strong randomness, there are still a large number of redundant points and inflection points in the final trajectory, which is more obvious in the more complex obstacles. These redundant points and inflection points will make the robot arm take a redundant track and unnecessary steering during the actual movement, which will seriously cause irreversible loss of the robot arm.

In order to eliminate redundant points and superfluous inflection points, the planned trajectory needs to be simplified. The specific simplification process is shown in Figure 13.

For an already planned trajectory, a direct connection is made from the starting point $X_{start}$ to the goal point $X_{goal}$, and if no collision occurs in the trajectory, this trajectory is taken as the final simplified trajectory. If a collision occurs, the trajectory is directly connected from $X_{start}$ to the previous node of $X_{goal}$, and if a collision occurs in this trajectory it is continued from $X_{start}$ to the previous node of this node. Instead, the node is used as the new parent node, and the above steps are repeated from that parent node until the parent node is directly connected to the goal point, representing the completion of the simplification of the trajectory.

### 4.2. Artificial Potential Field Improvement Strategies

#### 4.2.1. Rewriting the Potential Field Function

A typical problem of the traditional artificial potential field algorithm is that the force field is unreasonable, i.e., the problem of too small gravitational force and too large repulsive force occurs in some special locations [23]. For this situation above, the function of gravitational force and repulsive force to optimize are rewritten. The rewritten gravitational field function and gravitational force function are shown in Equations (10) and (11), respectively.
(10)$$U_{aat}(q)=\left\{\begin{array}{l} \frac{1}{2}K_{aat}\rho {\left(q,q_{goal}\right)}^{2},\;\rho (q,q_{goal})\leq d\\ dK_{aat}\rho (q,q_{goal})-\frac{1}{2}K_{aat}d^{2},\rho (q,q_{goal})>d \end{array}\right.$$
(11)$$F_{aat}(q)=-\nabla U_{aat}(q)=\left\{\begin{array}{l} -K_{aat}\rho (q,q_{goal}),\rho (q,q_{goal})\leq d\\ -dK_{aat}\frac{\partial \rho (q,q_{goal})}{\partial q},\rho (q,q_{goal})>d \end{array}\right.$$
where *d* is the distance factor, which sets a particular distance, and the rest of the parameters have the same meaning as before. According to this formula, when the position is farther away from the target point, the gravitational force received is reduced, and the magnitude of the reduction is distance-dependent.

#### 4.2.2. Add Target Points

One of the notable problems with the artificial potential field algorithm is its inability to reach the goal point, which is mainly due to the fact that the repulsive and gravitational forces exerted during the motion continue to change, resulting in zero combined force at a particular location. This causes the algorithm to mistakenly believe that it has reached the goal point and, therefore, stops moving, which ultimately leads to the planning algorithm failing to successfully reach the goal point [24,25].

In order to avoid the artificial potential field algorithm suggesting the target point is not reachable, in the case of a combined force of zero, the method of adding random target points can be used. Through the setting of random target points to break the original force balance, so that the combined force is not zero, continue to move to the target point. Random target setting can be used in order to ensure that it can continue to move at the same time. However, it is necessary to consider if the gravitational force is too large to cause a collision with the obstacles. To increase the principle of the target point in the current position on the obstacle with the largest force to take the right symmetry, the distance needs to be greater than the obstacle of the repulsive force range. Increasing the target point will allow it to continue to exist until it is out of the repulsive force range of the obstacle, as shown in Figure 14.

## 5. Obstacle Avoidance Trajectory Planning Algorithm Based on Improved Artificial Potential Field Fusion with Bi-RRT

Due to the existence of the local minimum problem, the trajectories planned by the improved artificial potential field algorithm are not optimal and have a large number of useless routes, while the improved Bi-RRT algorithm has fast planning speed and better global trajectories. In order to obtain a trajectory planning algorithm suitable for dynamic environments with better planning trajectories, the combination of the improved Bi-RRT algorithm and the improved artificial potential field algorithm is considered. The fusion algorithm is divided into two stages, namely static global trajectory planning and dynamic local trajectory planning. That is, the improved Bi-RRT algorithm is first utilized to carry out global trajectory planning in the static environment, and a globally better trajectory can be quickly obtained due to the short planning time and excellent planning route of the improved Bi-RRT algorithm. Subsequently, the improved artificial potential field algorithm is utilized to obtain dynamic obstacle information in real-time, and the global trajectory obtained in the static global trajectory planning stage is used to establish a guided potential field to avoid obstacles and converge to the global trajectory in real-time. The global planning trajectory of the bootstrap potential field is as follows:(12)$$U_{att}(q_{all})=K_{rrt}\rho {\left(q,q_{all}\right)}^{2}$$
where $\rho {\left(q,q_{all}\right)}^{2}$ is the square of the shortest distance between the current position and the global trajectory, and $K_{rrt}$ is the gravitational gain coefficient of the guiding potential field.

The gravitational potential field of the fusion algorithm consists of the gravitational potential field of the target point and the gravitational potential field of the global trajectory:(13)$$U_{att}=U_{att}(q_{goal})+U_{att}(q_{all})$$

The specific workflow is as follows:(1)Set the parameters of the banding robotic arm and obstacles, establish the initial position X_start_ and the target position X_goal_, and initialize the growth trees T1 and T2. Solve the workspace of the banding robotic arm according to the positive kinematics, regard the singular interval of the banding robotic arm as the obstacle interval, and calculate the feasible domain N of the banding robotic arm by combining the obstacle information;(2)Start alternating growth from the T1 tree or T2 tree as the current growing tree, generate a random factor P_r_, and calculate the minimum distance between the newest node of the current tree and the nearest obstacle d. Determine the direction of expansion according to P_r_, generate a new node X_new_, and check whether it is within the feasible domain N;(3)Envelope the robotic arm with material and obstacles and check the collision situation. If there is a collision, then go back to step 2. If there is no collision, then calculate the distance between the latest node of the two trees. If it is less than the value of the starting judgment distance ds, try to connect the two trees. If no collision occurs in the connected trajectory, then the planning trajectory is found, and the trajectory simplification should be carried out;(4)According to the simplified trajectory, the obstacle information can be obtained in real-time and the gravitational field and repulsive field function can be constructed. Judge whether the combined force is zero at that time. If it is zero, then add a target point and enter step (5). Instead, advance to the next point and enter step (6);(5)Advance to the next point and determine whether the current position escapes from the repulsive force range of the obstacle referenced by the incremental target point. If it does, remove the incremental target point and go back to step (4), and vice versa, repeat step (5);(6)Determine whether the target point is reached. If so, end the algorithm and find the complete planning trajectory; otherwise, return to step (4) to continue the execution.

The flow chart of the obstacle avoidance trajectory planning algorithm based on the fusion of improved artificial potential field and Bi-RRT is shown in Figure 15.

## 6. Experimental Validation

### 6.1. Theoretical Validation

When designing the artificial potential field, the setting and adjustment of parameters are very important for achieving good obstacle avoidance. The parameter settings need to be reasonably selected according to the specific application scene.

In order to compare and analyze the improvement of the fusion algorithm, the fusion algorithm and the improved artificial potential field algorithm are compared in MATLAB. The parameters of the algorithm are shown in Table 2.

A circular obstacle with the center of the circle [4, 4], [4, 6], [6, 4], a radius of 0.5 m is set in the environment. The final simulation result is shown in Figure 16.

According to the improved artificial potential field roadmap in Figure 16a, it can be found that the algorithm is not within the obstacle repulsive range at the beginning and is only affected by the gravitational force and the direction of the combined force points to the endpoint; the beginning trajectory is close to a straight line. When entering the repulsive range of the obstacle, there is a local minimum, reciprocating movement in a certain area, resulting in trajectory redundancy. In the planning effect diagram of different algorithms in Figure 17, the RRT algorithm performs the worst, with a large number of transitions during the growth of its nodes, resulting in a long expansion time. The RRT* algorithm has progressive optimization characteristics, which makes the trajectory length relatively shorter and smoother overall, but it also means that its convergence time will increase. The Bi-RRT trajectory is similar to the RRT, and there are more inflection points in the nodes, but the algorithm convergence time is faster. The fusion algorithm, on the other hand, outperforms the comparison algorithm in terms of trajectory length and trajectory quality.

### 6.2. Simulation Validation

The experiments were performed using a virtual machine with the Ubuntu 18.04 operating system, with a melodic version of the ROS platform installed. For hardware, a PC (CPU i5-12600KF @3.70 GHz, RAM 32G, GPU RTX 4060TI 16G) was used to support the experiments.

The drilling and anchoring robot system model was established through Solidworks, which mainly includes the robotic arm with the material, a material magazine, a drilling machine, and a roof plate. It was imported into the ROS platform to build the working environment of the coal mine roadway excavation support. Simulate the trajectory planning algorithm in the ROS platform, write the corresponding code of the algorithm, and build the corresponding robot arm trajectory planning library. Keeping the rest of the parameters the same, simulate the support process of a drill hole and compare and analyze the planning results of different algorithms.

The following is a simplified step-by-step procedure for the drilling and anchoring robotic system to carry out the support work of coal mine tunnel excavation:(1)Robotic arm for loading and unloading drill rods for drilling rig;(2)Robotic arm facing the drilling hole and loading medical roll;(3)Robotic arm facing the drilling rig for loading anchor rods.

The figure above shows the simulation results of robotic arm motion planning, in which the yellow surface represents the motion trajectory of the end of the robotic arm from the starting position to the final position. In the figure, the path planning carried out by the robotic arm in the state of clamping the drill pipe, the medicine roll, and the anchor, respectively, still maintains the vertical state of clamping objects, which is in line with the attitude requirements of the drilling and anchoring robotic system in carrying out the work of tunnel excavation support in coal mines. According to the trajectory diagrams from Figure 18, Figure 19, Figure 20 and Figure 21, it can be found that the fusion algorithm plans the trajectory without any collision during the movement process; the overall process is smooth, and the trajectory is in the feasible domain of the robotic arm.

Table 3 shows the simulation data of different algorithms. Comparing with Table 3, it can be found that the longest planning time of the RRT algorithm is not ideal. Although Bi-RRT has a shorter time, the trajectory is longer, and there are more inflection points, and the RRT* algorithm has a short trajectory length due to its asymptotically optimal characteristics. However, the planning time of each time is up to the maximum, and this paper’s algorithm is optimal in terms of both the speed and quality of the planning. The planning time is up to 69.8% compared to the RRT and up to 45.3% compared to Bi-RRT. The trajectory length is 29.8% compared to Bi-RRT and 13.4% compared to RRT*. The fusion algorithm has the highest success rate in the planning process, and the number of iterations is relatively shortest.

## 7. Conclusions

In the coal mine tunnel drilling and anchoring support process, when the robotic arm cooperates with the drilling machine to load and unload the drilling and anchoring materials, it is difficult to plan the trajectory in the dynamic environment. An obstacle avoidance trajectory planning algorithm based on the fusion of the improved artificial potential field and Bi-RRT is proposed. The global trajectory planning in a static environment is carried out by improving the Bi-RRT algorithm, and then the guiding potential field is established based on the optimized trajectory and real-time obstacle information; the final trajectory is derived using the improved artificial potential field method. Through Matlab simulation analysis, it is verified that the fusion algorithm has a better path-planning effect than the traditional algorithm and avoids the local minimum solution. The feasibility of the algorithm in the drilling and anchoring robot system is verified through the ROS platform Rviz simulation.

In order to further study the adaptability of the relevant algorithm in the complex environment of a coal mine underground, the drilling and anchoring robot system platform will be built in future work to simulate the real permanent support scene and improve the robustness of the algorithm.

## Figures and Tables

**Figure 1 sensors-24-06866-f001:**
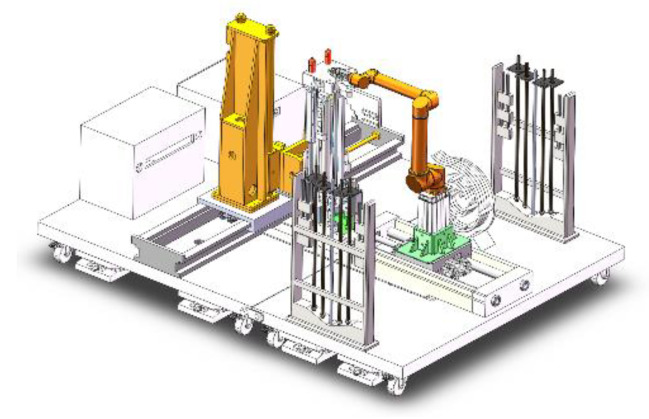
Structure of anchor drilling robot system.

**Figure 2 sensors-24-06866-f002:**
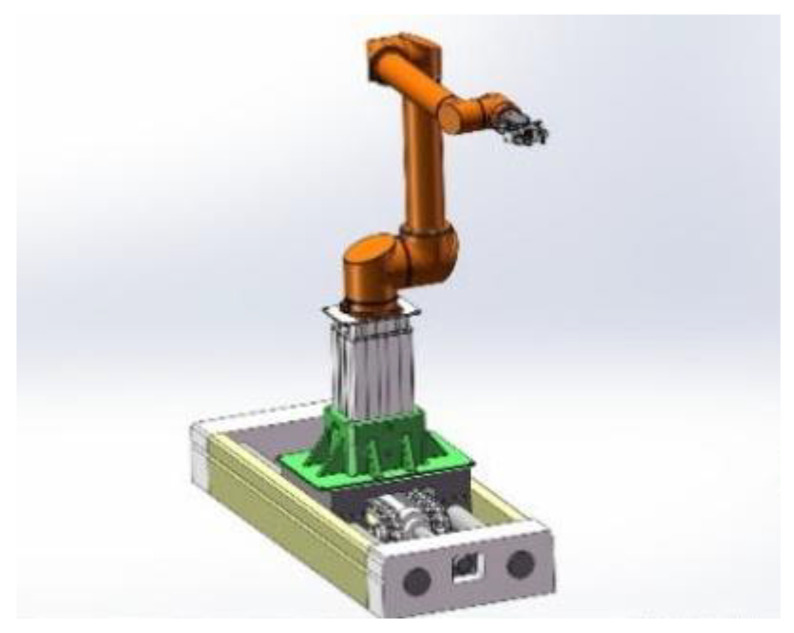
Overall structure of anchor drilling robot arm.

**Figure 3 sensors-24-06866-f003:**
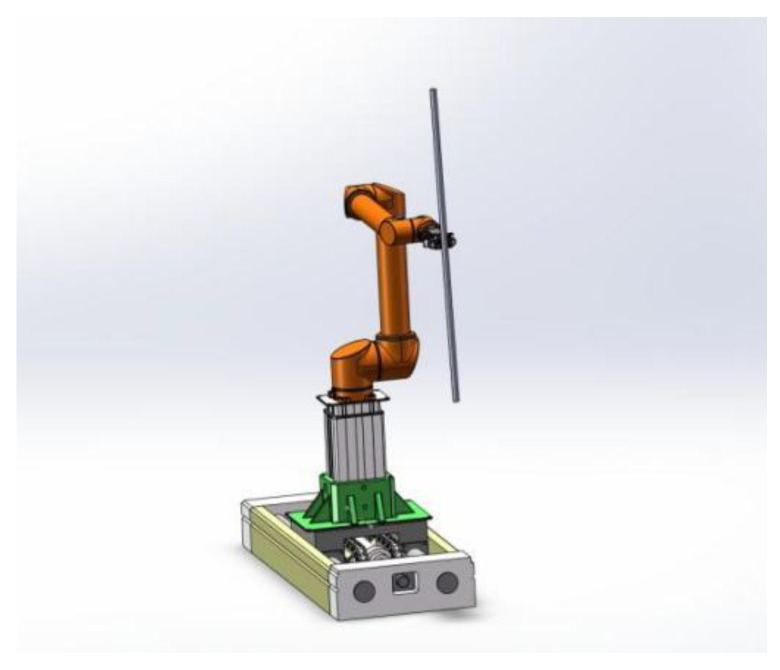
Schematic diagram of a mechanical arm with material.

**Figure 4 sensors-24-06866-f004:**
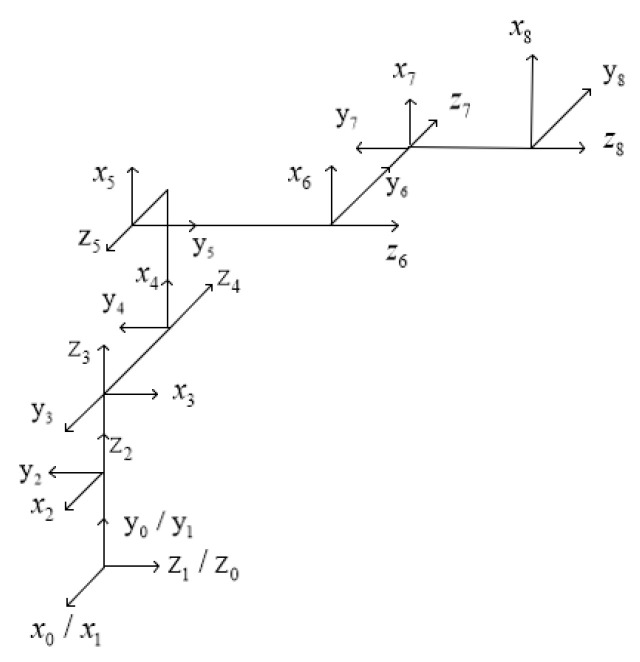
Coordinate diagram of connecting rod of the cooperative manipulator.

**Figure 5 sensors-24-06866-f005:**
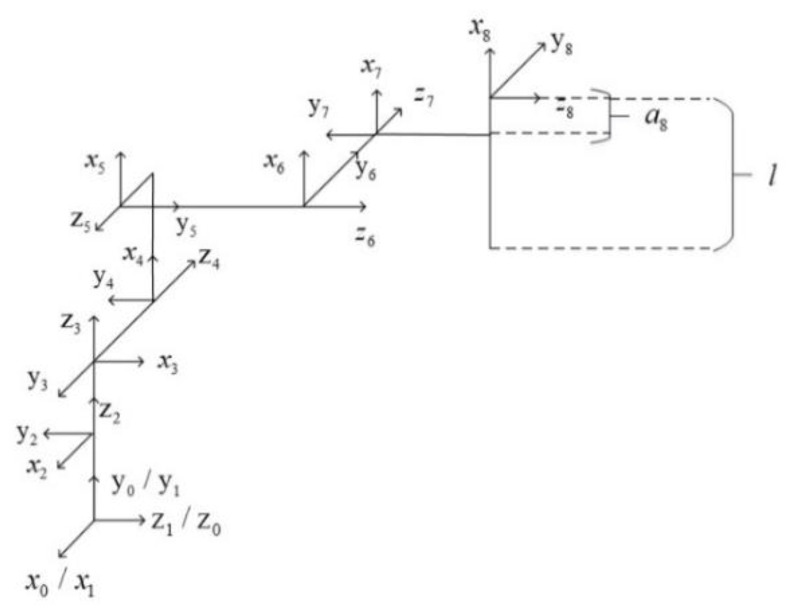
Drawing of connecting rod coordinate system of conveyor arm.

**Figure 6 sensors-24-06866-f006:**
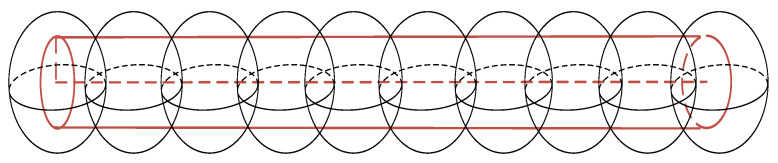
Simplified model of obstacles.

**Figure 7 sensors-24-06866-f007:**
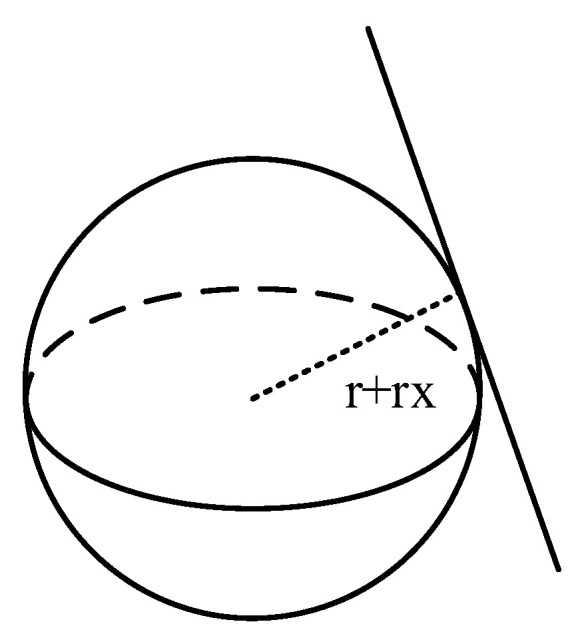
Collision of connecting rod with obstacle.

**Figure 8 sensors-24-06866-f008:**
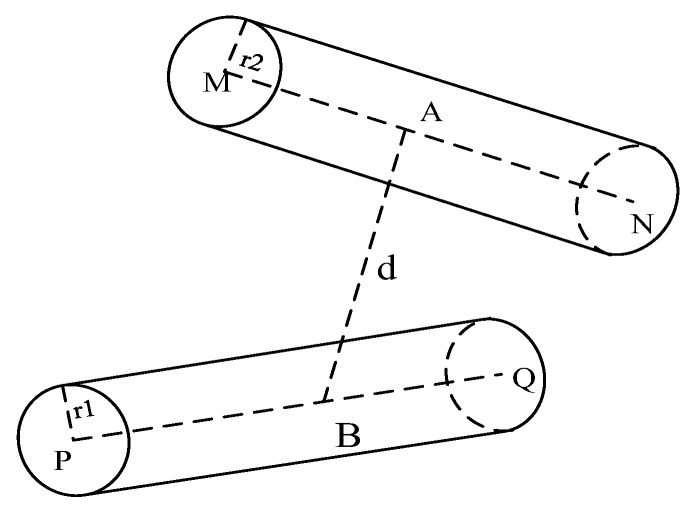
The collision between rods.

**Figure 9 sensors-24-06866-f009:**
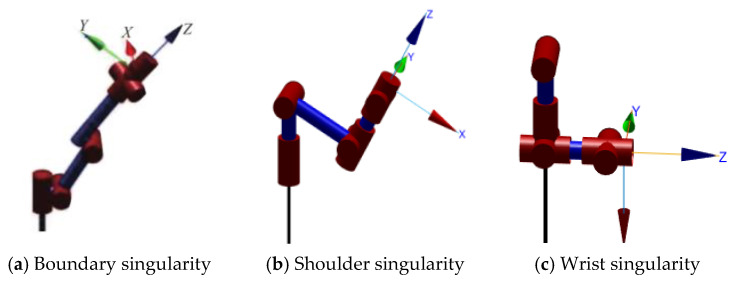
Different singular bit patterns.

**Figure 10 sensors-24-06866-f010:**
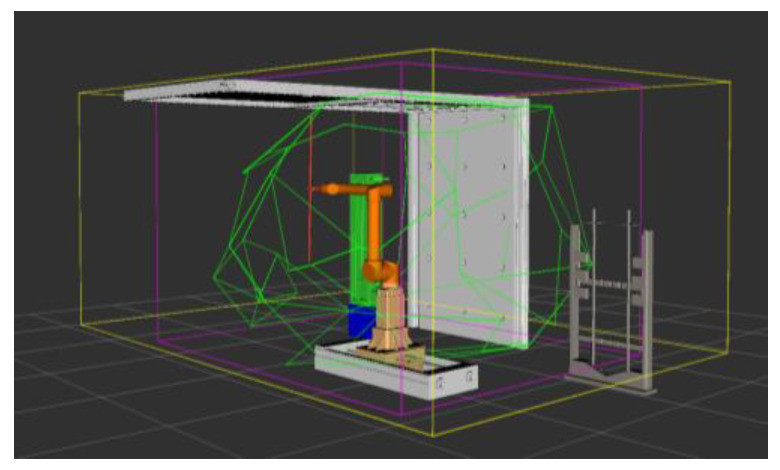
Feasible domain of the robotic arm.

**Figure 11 sensors-24-06866-f011:**
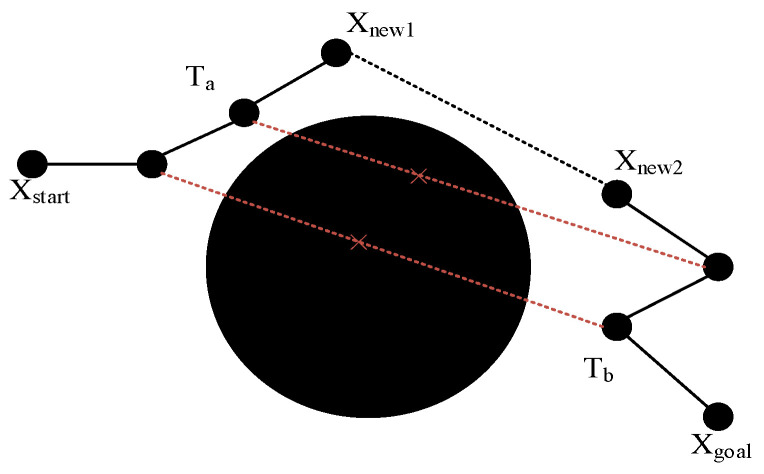
Connection Judgment Strategy.

**Figure 12 sensors-24-06866-f012:**
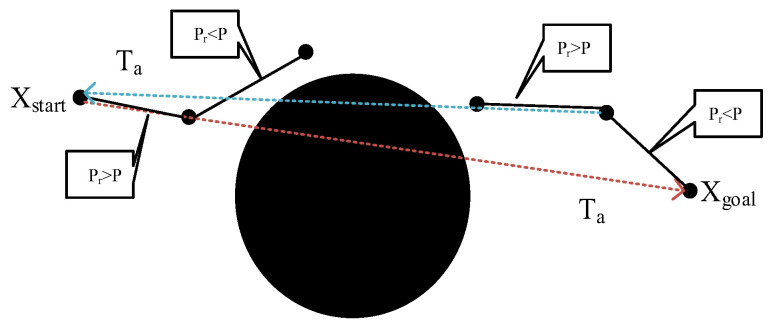
Goal Oriented Strategy.

**Figure 13 sensors-24-06866-f013:**
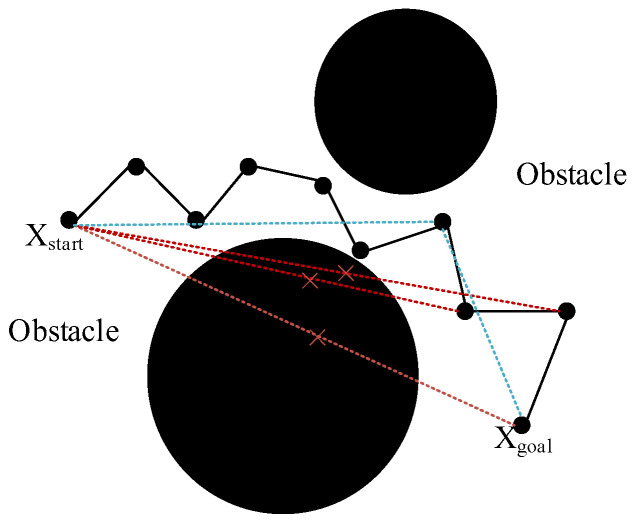
Trajectory simplification strategy.

**Figure 14 sensors-24-06866-f014:**
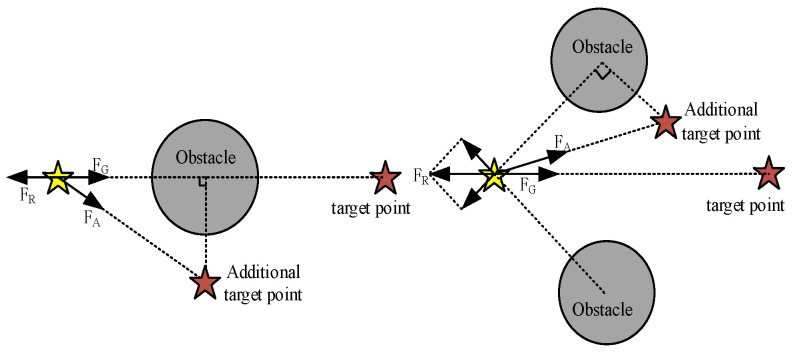
Add Target Points.

**Figure 15 sensors-24-06866-f015:**
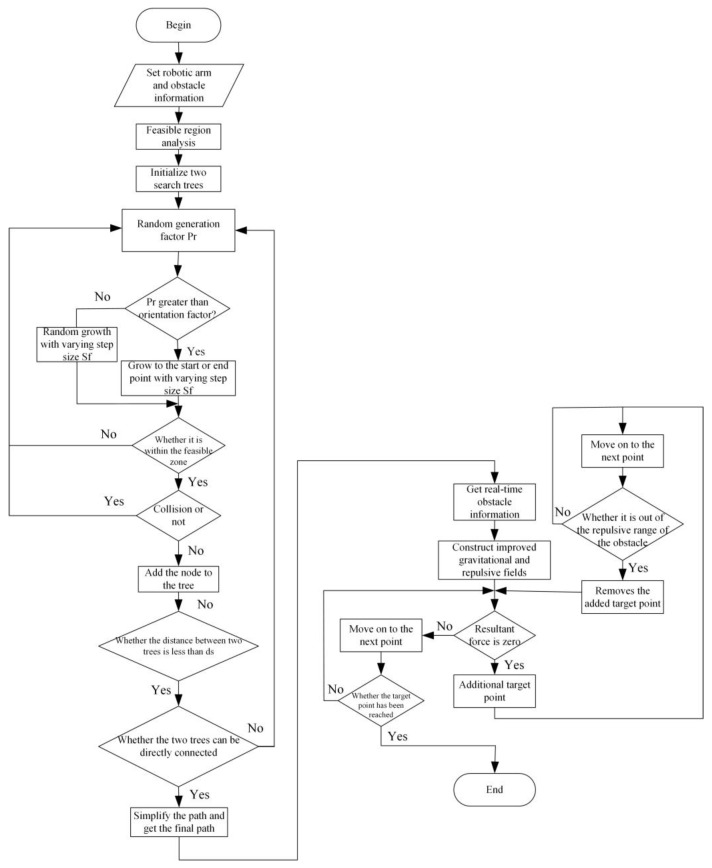
Based on the flow chart of the fusion algorithm of improved artificial potential field and Bi-RRT.

**Figure 16 sensors-24-06866-f016:**
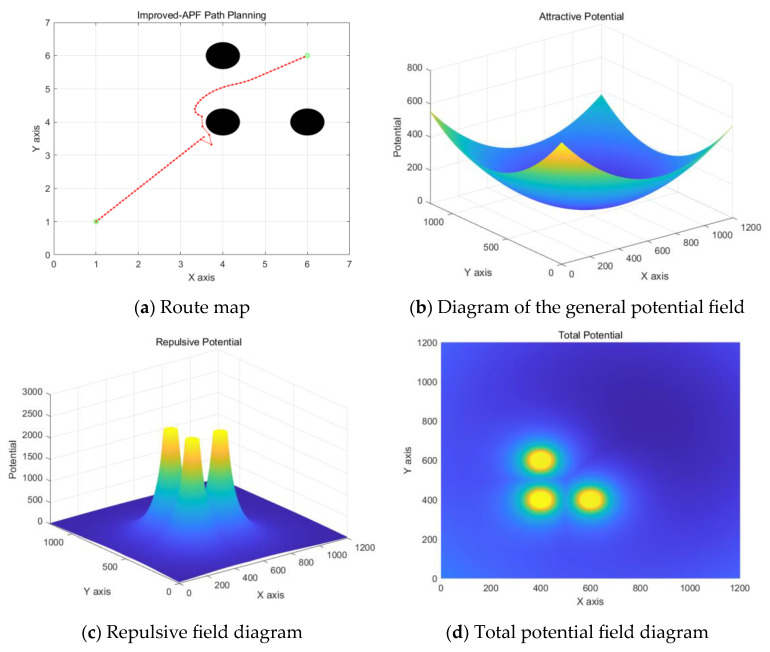
Improved artificial potential field simulation diagram.

**Figure 17 sensors-24-06866-f017:**
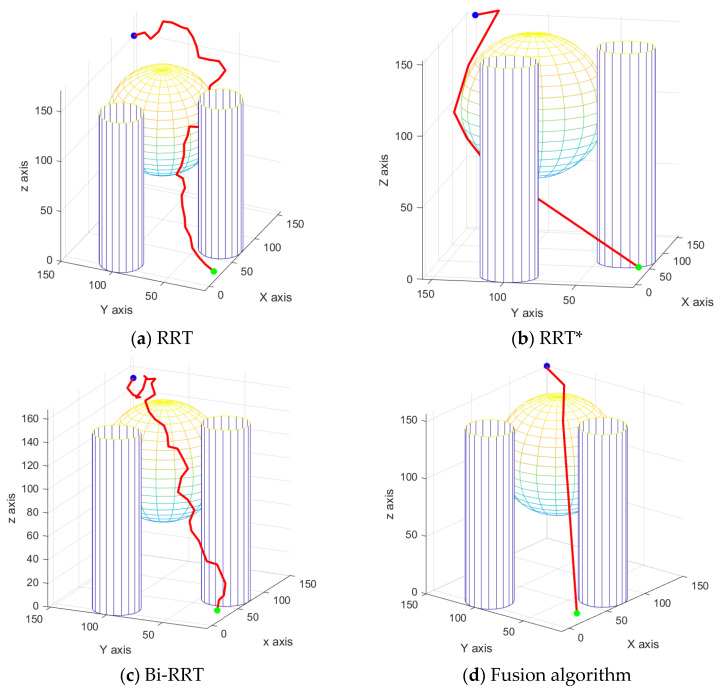
Different algorithm planning effect.

**Figure 18 sensors-24-06866-f018:**
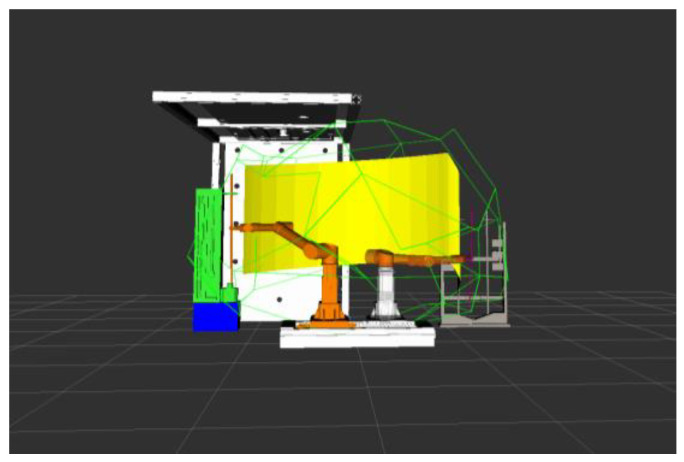
Grabbing the drilling rods and loading them into the drilling rig.

**Figure 19 sensors-24-06866-f019:**
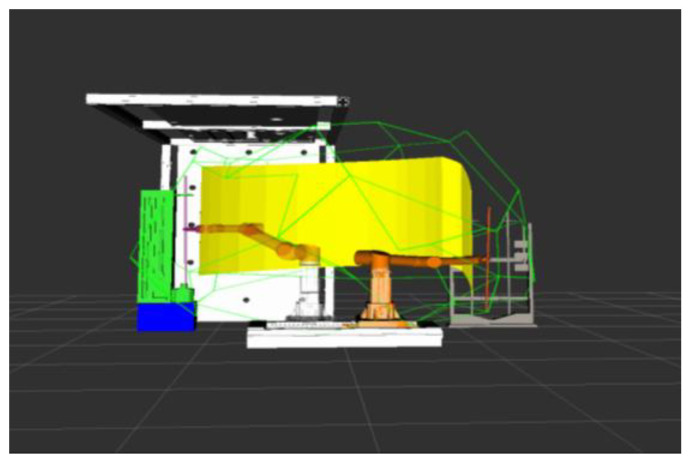
Remove the drill rod from the drilling rig and bring it back to the material depot.

**Figure 20 sensors-24-06866-f020:**
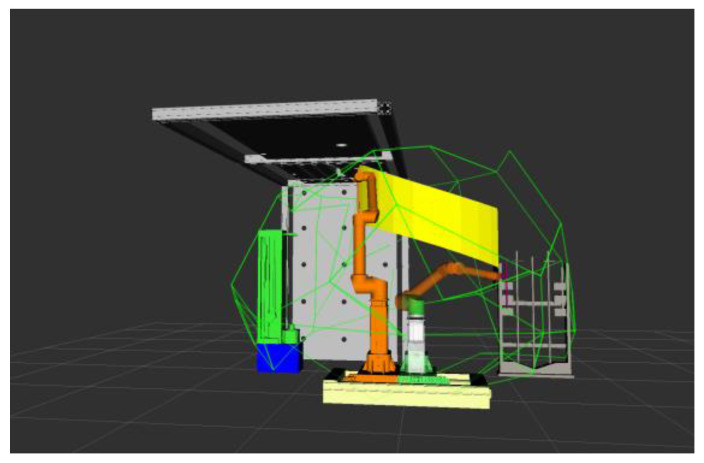
Clamping and loading of medical rolls into top plate drill holes.

**Figure 21 sensors-24-06866-f021:**
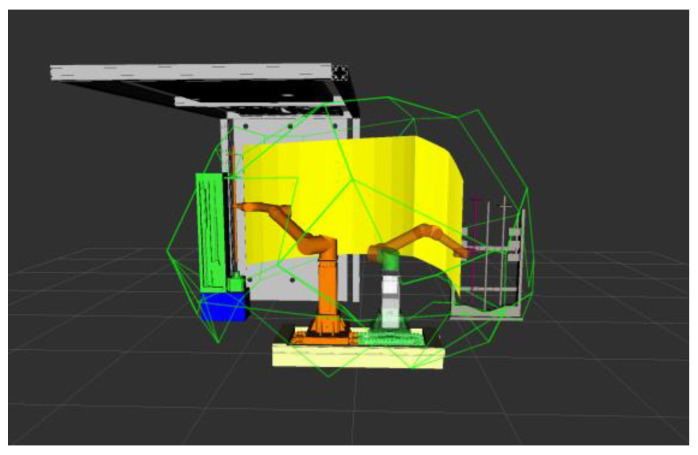
Grabbing anchor rods for loading into the drilling rig.

**Table 1 sensors-24-06866-t001:** D-H parameters of drilling anchor robotic arm.

** *i* **	$\alpha_{i-1}$	$a_{i-1}$	$d_{i}$	$\theta_{i}$
1	0	0	$d_{1}$	0
2	90	0	$d_{2}$	0
3	0	0	10	$\theta_{3}$
4	90	10	0	$\theta_{4}$
5	−90	60	10	$\theta_{5}$
6	90	0	60	$\theta_{6}$
7	90	0	10	$\theta_{7}$
8	−90	$a_{8}$	10	$\theta_{8}$

**Table 2 sensors-24-06866-t002:** Algorithm parameters setting.

Parameter	Value
$\mathrm{Gravitational}\;\mathrm{coefficient}\;K_{a}$	28
$\mathrm{Repulsion}\;\mathrm{coefficient}\;K_{r}$	15
$\mathrm{Repulsive}\;\mathrm{force}\;\mathrm{influence}\;\mathrm{range}\;d_{0}$/m	2
Step size/m	0.1
Maximum number of iterations	200

**Table 3 sensors-24-06866-t003:** Simulation data of different algorithms.

**Algorithm**	**Task 1**				**Algorithm**	**Task 2**			
	**Planning Time/s**	**Trajectory Length/mm**	**Success Rate**	**Iteration Frequency**		**Planning Time/s**	**Trajectory Length/mm**	**Success Rate**	**Iteration Frequency**
RRT	0.57	8770	56%	181	RRT	0.33	3580	62%	146
Bi-RRT	0.23	5352	78%	156	Bi-RRT	0.15	2782	86%	132
RRT*	5	5236	50%	124	RRT*	5	2460	66%	135
Ours	0.11	4050	100%	76	Ours	0.07	2320	100%	62
**Algorithm**	**Task 3**				**Algorithm**	**Total**			
	**Planning Time/s**	**Trajectory Length/mm**	**Success Rate**	**Iteration Frequency**		**Planning Time/s**	**Trajectory Length/mm**	**Success Rate**	**Iteration Frequency**
RRT	0.46	6850	64%	166	RRT	1.36	19,130	60.7%	164
Bi-RRT	0.37	6436	88%	168	Bi-RRT	0.75	14,570	84%	152
RRT*	5	4112	58%	148	RRT*	15	11,808	58%	135
Ours	0.23	3850	100%	102	Ours	0.41	10,220	100%	80

## Data Availability

The data presented in this study are available on request from the author.

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
