# Peer review of "Research on Obstacle-Avoidance Trajectory Planning for Drill and Anchor Materials Handling by a Mechanical Arm on a Coal Mine Drilling and Anchoring Robot"

_sensors, 2024, doi:10.3390/s24216866_

Round 1

Reviewer 1 Report

Comments and Suggestions for Authors

The manuscript is too long and with much unnecessary content, please remove some well-known method and description, just put your novel idea and method in the manuscript.

1. The abstract is too long and superfluous for readers, remove some detailed description in the abstract, such as the D-H method and Monte Carlo method, etc. Just keep the novel idea and the key features of your new approaches in the abstract. In addition, suggested changes to the abstract, especially adding some numerical results to validate the original contribution of this paper.

2. Remove some of the content in the methodology section, e.g., the section "2.1 Kinematics modeling of material-carrying robotic arm", because D-H method is a well-known method, and the robot arm you used are also well-known robot arm, do not introduce it in detail.

3. Remove the Figure 13 and the corresponding introduction for the algorithm's steps, this method is also well-known, keep Figure 14 is enough. This manuscript is for research papers, not for scientific reports.

4. For the experiments, there is only simulation on the MATLAB and ROS, it is better to conduct some experiments on the real robot arms, even with some small robot arms in the lab is ok. 

5. Please try to describe the experiment in a more systematic way and explain the results and analysis in detail.

6. The contribution of this paper is not well explained, the contributions could be more precise.

Comments on the Quality of English Language

1. Enhancement the language of the paper. In particular, it is necessary to streamline the descriptions in the paper, which are currently too wordy.

2. There are some typos, e.g., in the first paragraph of Section 2.2, "For regular irregular obstacles ..."; below equation (24), "ara \lambda, \mu, The ...", there's an extra space. etc. Please go through the manuscript.

Author Response

The manuscript is too long and with much unnecessary content, please remove some well-known method and description, just put your novel idea and method in the manuscript.

Point 1: The abstract is too long and superfluous for readers, remove some detailed description in the abstract, such as the D-H method and Monte Carlo method, etc. Just keep the novel idea and the key features of your new approaches in the abstract. In addition, suggested changes to the abstract, especially adding some numerical results to validate the original contribution of this paper.

Response 1: Thanks to the careful work of experts. Following expert advice, we have revised the abstract to remove the description of the known method details, while adding the overall simulation results to further reflect the contribution of this paper. Changes have been highlighted in red in the abstract.

Point 2: Remove some of the content in the methodology section, e.g., the section "2.1 Kinematics modeling of material-carrying robotic arm", because D-H method is a well-known method, and the robot arm you used are also well-known robot arm, do not introduce it in detail.

Response 2: Thanks for the experts' suggestions. According to the expert opinions, we have redescribed the structure and modeling idea of the material-carrying robotic arm mentioned in the paper, and simplified the derivation process of the kinematic modeling theory of the material-carrying robotic arm. The modified part has been marked in red in section 2.1 on page 4.

Point 3: Remove the Figure 13 and the corresponding introduction for the algorithm's steps, this method is also well-known, keep Figure 14 is enough. This manuscript is for research papers, not for scientific reports.

Response 3: Thanks for the experts' suggestions. According to the expert opinions, we have re-condensed the principle of fusion algorithm of improved artificial potential field and Bi-RRT, and at the same time re-integrated the contents of Figure 13 and Figure 14 in the original text, which as shown in Figure 15. The changes are highlighted in red on pages 12-13.

Point 4: For the experiments, there is only simulation on the MATLAB and ROS, it is better to conduct some experiments on the real robot arms, even with some small robot arms in the lab is ok.

Response 4: Thanks for the experts' suggestions. Experiments on real robotic arms can verify the effectiveness of the algorithm. However, due to the limited conditions of this experiment, in order to further verify the effectiveness of the relevant algorithms in this paper, the anchoring robot system platform will be built for testing and verification in the future work. We have made a supplementary description of this part in the outlook part of the conclusion, and the modified part has been marked in red font. Thank you again for your understanding.

Point 5: Please try to describe the experiment in a more systematic way and explain the results and analysis in detail.

Response 5: Thanks for the experts' suggestions. According to the expert opinion, we have redescribed the experimental part and analyzed the experimental results in detail, as shown in Table 3. The changes are highlighted in red on page 17.

Table 3. Simulation data of different algorithms.

Algorithm

Task 1

Algorithm

Task 2

Planning time/s

Trajectory length/mm

Success rate

Iteration

frequency

Planning time/s

Trajectory length/mm

Success rate

Iteration

frequency

RRT

0.57

8770

56%

181

RRT

0.33

3580

62%

146

Bi-RRT

0.23

5352

78%

156

Bi-RRT

0.15

2782

86%

132

RRT*

5

5236

50%

124

RRT*

5

2460

66%

135

Ours

0.11

4050

100%

76

Ours

0.07

2320

100%

62

Algorithm

Task 3

Algorithm

Total

Planning time/s

Trajectory length/mm

Success rate

Iteration

frequency

Planning time/s

Trajectory length/mm

Success rate

Iteration

frequency

RRT

0.46

6850

64%

166

RRT

1.36

19130

60.7%

164

Bi-RRT

0.37

6436

88%

168

Bi-RRT

0.75

14570

84%

152

RRT*

5

4112

58%

148

RRT*

15

11808

58%

135

Ours

0.23

3850

100%

102

Ours

0.41

10220

100%

80

Point 6: The contribution of this paper is not well explained, the contributions could be more precise.

Response 6: Thanks for the experts' suggestions. Following expert advice, we have redescribed the contributions in this article to make them more accurate. The changes are highlighted in red at the conclusions of page 18.

Comments on the Quality of English Language

Point 7: Enhancement the language of the paper. In particular, it is necessary to streamline the descriptions in the paper, which are currently too wordy.

Response 7: Thanks for the careful work. According to the expert opinion, we have carefully checked the language description of the full text and simplified the excessively lengthy description in the article.

Point 8: There are some typos, e.g., in the first paragraph of Section 2.2, "For regular irregular obstacles ..."; below equation (24), "ara \lambda, \mu, The ...", there's an extra space. etc. Please go through the manuscript.

Response 8: Thanks for the careful work. According to expert opinions, we have carefully checked the full text and corrected some erroneous descriptions in the article.

Reviewer 2 Report

Comments and Suggestions for Authors

Please note the following comments:

    - Include a detailed section explaining the derivations and assumptions of the kinematic model and collision detection methods.

    - Provide further explanation of how specific parameters for the algorithms used, such as steering factors and step sizes, were determined.

    - Detail the experimental setup, including specific hardware and software configurations used in the ROS and MATLAB simulations.

    - Add more information about the evaluation criteria and matrices used to compare the performance of different algorithms.

    - Expand the results section to include more detailed comparative analysis between the proposed and existing algorithms.

    - Discuss current research limitations and suggest future directions for further studies.

Author Response

Point 1: Include a detailed section explaining the derivations and assumptions of the kinematic model and collision detection methods.

Response 1: Thanks to the careful work of experts. In this paper, the connecting rod and rod-shaped material of the manipulator are enveloped by a cylinder, and the rod-shaped obstacle is also enveloped by a series of spheres with smaller radius. The spatial pose information of the connecting rod is obtained by solving the kinematics model, and the collision is judged according to the radius information of the envelope cylinder or sphere. We have introduced the principle of collision detection of the charging arm in Section 2.2, which has been highlighted in red on page 6.

Point 2: Provide further explanation of how specific parameters for the algorithms used, such as steering factors and step sizes, were determined.

Response 2: Thanks for the experts' suggestions. We have added the theoretical basis and principle of parameter setting in the paper. After several experiments, the results show that when the parameters are set as shown in Table 2, the motion trajectory is the smoothest and the planning time is the shortest. The changes are highlighted in red on page 13, section 5.1.

Point 3: Detail the experimental setup, including specific hardware and software configurations used in the ROS and MATLAB simulations.

Response 3: Thanks for the experts' suggestions. According to the expert advice, we have added the specific hardware and software configuration information in Section 5.2 of the article, and the added part is marked in red font on page 15. Specific details are as follows:

The configuration information is as follows: The ROS platform melodic is installed on a VM running the Ubuntu 18.04 operating system. In terms of hardware, PC (CPU i5-12600KF@ 3.70GHz, RAM 32G, GPU RTX 4060TI 16G) was used to support the experiment. 

Point 4: Add more information about the evaluation criteria and matrices used to compare the performance of different algorithms.

Response 4: Thanks for the experts' suggestions. According to the expert opinion, we have added the evaluation criteria of success rate and number of iterations to further verify the advantages of the algorithm in this paper. The supplementary experimental results are shown in Table 3, and the specific modified part is marked in red font in Table 3 on page 17.

Table 3. Simulation data of different algorithms.

Algorithm

Task 1

Algorithm

Task 2

Planning time/s

Trajectory length/mm

Success rate

Iteration

frequency

Planning time/s

Trajectory length/mm

Success rate

Iteration

frequency

RRT

0.57

8770

56%

181

RRT

0.33

3580

62%

146

Bi-RRT

0.23

5352

78%

156

Bi-RRT

0.15

2782

86%

132

RRT*

5

5236

50%

124

RRT*

5

2460

66%

135

Ours

0.11

4050

100%

76

Ours

0.07

2320

100%

62

Algorithm

Task 3

Algorithm

Total

Planning time/s

Trajectory length/mm

Success rate

Iteration

frequency

Planning time/s

Trajectory length/mm

Success rate

Iteration

frequency

RRT

0.46

6850

64%

166

RRT

1.36

19130

60.7%

164

Bi-RRT

0.37

6436

88%

168

Bi-RRT

0.75

14570

84%

152

RRT*

5

4112

58%

148

RRT*

15

11808

58%

135

Ours

0.23

3850

100%

102

Ours

0.41

10220

100%

80

Point 5: Expand the results section to include more detailed comparative analysis between the proposed and existing algorithms.

Response 5: Thanks for the experts' suggestions. According to expert opinions, we have systematically described the experimental process and supplemented the comparative experimental results of other algorithms, as shown in Figure 16 and 17. The changes are highlighted in red on pages 14-15, Section 5.1.

Point 6: Discuss current research limitations and suggest future directions for further studies.

Response 6: Thanks for the experts' suggestions. Although the proposed algorithm achieves good results in trajectory planning time, the real-time response performance of the proposed algorithm needs to be further improved in the face of complex environmental changes. In the future work, we will focus on the response speed of the algorithm in the face of complex environmental changes.
